

# Performance comparison between CERES-MODIS and OMI in retrieving SSA across diverse aerosol regimes

Archana Devi[1], Sreedharan K Satheesh[1,2], Jayaraman Srinivasan[2]

[1] Centre for Atmospheric and Oceanic Sciences, Indian Institute of Science, Bengaluru, India
[2] Divecha Centre for Climate Change, Indian Institute of Science, Bengaluru, India

*Correspondence to:* Archana Devi (archana.shiva13@gmail.com)

## Abstract

A comparative evaluation of aerosol single scattering albedo (SSA) retrieved from CERES-MODIS and OMI satellite instruments is presented across diverse aerosol environments. The analysis focuses on seven key regions: biomass burning areas (Amazon and South African Congo forests), clean and polluted oceanic zones (South Pacific, Arabian Sea, and Bay of

Bengal), clean and polluted land areas (North America, Europe, Indo-Gangetic Plain, and Eastern China), and the Sahara Desert. Monthly aerosol type concentrations from MERRA-2 reanalysis data are used to investigate the seasonal variation in aerosol loading and composition. Results show that CERES-MODIS consistently captures SSA variability more effectively than OMI, particularly in regions dominated by absorbing aerosols such as black

carbon. In biomass burning regions, CERES-MODIS displays a strong negative correlation between fire count and SSA, unlike OMI, which shows weaker or negligible correlations. Over clean regions, both instruments perform comparably, while over polluted zones and deserts, CERES-MODIS demonstrates greater sensitivity to aerosol type and seasonal trends. The findings highlight the relative strengths and limitations of both algorithms in aerosol

monitoring under diverse atmospheric conditions.





# 1 Introduction

Aerosols significantly influence Earth's radiation budget by directly scattering and absorbing solar radiation and indirectly affecting cloud properties and lifetimes. The magnitude and direction of these aerosol-radiation interactions depend critically on aerosol optical properties, particularly single scattering albedo (SSA), which quantifies the relative contributions of scattering versus absorption. Accurate estimation of SSA is therefore essential for assessing aerosol radiative forcing and its implications for climate change (Haywood and Boucher, 2000; IPCC, 2021). Satellite-based observations provide large-scale, long-term monitoring of aerosol optical properties (Kahn et al., 2007).

Satellite-based retrievals of SSA have been developed using several instruments, each with its own unique observational capabilities. The Ozone Monitoring Instrument (OMI) provides SSA estimates by exploiting its wide spectral coverage in the UV and visible regions, which is particularly useful for detecting absorbing aerosols such as dust and smoke (Schoeberl et al., 2006; Torres et al., 1998, 2005, 2007). The PARASOL (Polarization and Anisotropy of Reflectances for Atmospheric Sciences coupled with Observations from a Lidar) retrieves aerosol properties, including SSA, through multi-angle and multispectral photopolarimetric observations acquired at up to 16 different viewing directions across nine spectral channels between 0.44 and 1.02 μm (Chen et al., 2020; Lacagnina et al., 2015; Chen et al., 2022). By combining directional, spectral, and polarization information, the measurements place strong constraints on aerosol load and microphysical characteristics (Lacagnina et al., 2017; Hasekamp et al., 2024). Unfortunately, the PARASOL mission ended on 18 December 2013. Looking ahead, the PACE (Plankton, Aerosol, Cloud, ocean Ecosystem) mission is expected to significantly advance SSA retrievals by combining hyperspectral radiometry with polarimetry, enabling more accurate characterization of aerosol absorption over both land and ocean



(Hasekamp et al., 2019; Remer et al., 2019; Fu et al., 2025). Together, these efforts represent a progression of satellite remote sensing towards improved global monitoring of aerosol absorption property.

The work by Devi and Satheesh (2022) developed an aerosol SSA retrieval method by combining CERES-MODIS (Clouds and the Earth's Radiant Energy System) (Moderate Resolution Imaging Spectroradiometer) observations, and demonstrated its capability to retrieve SSA and capture it's spatial and seasonal variability. The CERES-MODIS method for SSA retrieval excels at directly determining SSA in the visible spectrum (550 nm) by synergistically combining CERES and MODIS data, providing crucial input for climate models and allowing for comprehensive global coverage of monthly or seasonal SSA maps. In contrast, OMI (Ozone Monitoring Instrument) SSA leverages UV wavelengths (330-400 nm) to retrieve SSA though it requires extrapolation to visible wavelengths.

This study extends the work of Devi and Satheesh (2022) by performing a systematic evaluation of SSA retrieved from CERES-MODIS and OMI across a diverse range of environment regimes. Specifically, we focus on six major environmental contexts: biomass burning regions (Amazon and South Africa), polluted and clean land regions (e.g., Indo-Gangetic Plain, North America), polluted and clean oceanic regions (Arabian Sea, South Pacific), and deserts (Sahara). Each of these regions represents distinct aerosol sources and compositions—ranging from black carbon-rich smoke plumes to dust-dominated atmospheres—providing an ideal framework to assess the robustness and sensitivity of satellite SSA retrievals under varying conditions.

The primary objective is to assess the ability of each algorithm to detect variations in absorbing aerosols under different atmospheric conditions. Special emphasis is placed on evaluating the



relationship between fire activity and SSA in biomass burning regions, where an increase in absorbing aerosols is expected to correspond with a reduction in SSA.

To support this analysis, monthly aerosol type concentrations are obtained from NASA's MERRA-2 reanalysis dataset (Gelaro et al., 2017), allowing for a detailed examination of

seasonal aerosol loading. This includes the decomposition of aerosol optical depth (AOD) by aerosol type (such as organic carbon, black carbon, sea salt, sulphates, and dust) across all selected regions. Seasonal SSA values retrieved from CERES-MODIS and OMI are compared to evaluate each algorithm's sensitivity to aerosol composition and variability. By jointly analyzing SSA and aerosol type AOD, this study provides insight into the performance,

reliability, and limitations of the two satellite products under diverse aerosol regimes and meteorological conditions. The findings serve to guide the selection of appropriate aerosol datasets for climate studies, air quality monitoring, and radiative forcing assessments.

## 2 Data Used

### 2.1 CERES-MODIS SSA

The CERES-MODIS SSA algorithm (Devi and Satheesh, 2022) is based on the critical optical depth ($\tau_c$) method, originally introduced by Satheesh & Srinivasan (2005). To implement this globally, the method uses flux (Top of atmosphere and surface) data from CERES, together with aerosol optical depth (AOD) from MODIS at 550 nm. The methodology involves 3 steps: (1) Computing $\tau_c$: $\Delta\alpha$ is the difference between surface and TOA albedo. A linear regression is

performed between $\Delta\alpha$ and AOD. The x-intercept of that regression is taken as $\tau_c$ (2) Look up tables (LUTs) were generated using radiative transfer simulations (3) An inverse look up operation is performed on the LUTs to invert those LUTs to retrieve SSA that corresponds to the estimated $\tau_c$. The retrieval is done globally, producing monthly or seasonal SSA maps (550 nm), with SSA retrieved per 1° grid.



The total uncertainty in retrieved SSA is estimated to be about ±0.044 (at 550 nm). The largest contributor to uncertainty is variation / error in surface albedo (±0.03 SSA error). The method was validated by comparing the retrieved SSA with in situ / airborne measurements: three aircraft field campaigns over India and adjacent oceans. Most of the SSA estimates from

CERES-MODIS lie within an absolute difference of ±0.03 of the aircraft measurements. Also, comparisons were performed with ground-based AERONET SSA and overall RMSE was ~0.026, and the agreement was acceptable within the combined uncertainties of both datasets.

CERES-MODIS SSA has the advantage that it retrieves SSA directly in the visible wavelength (550 nm). Whereas the most widely used OMI retrieves SSA in the UV and extrapolates to the

visible wavelengths. Compared to OMI, CERES-MODIS has better global data coverage with fewer data gaps. It distinctly captures the seasonal variation in SSA values and provides monthly and seasonal SSA datasets (Devi and Satheesh, 2022). Disadvantage is that it requires sufficient variation in the AOD values over the region of retrieval.

## 2.2 OMI SSA

The OMI, aboard the EOS-Aura satellite launched in July 2004, is a high-spectral-resolution spectrograph that measures upwelling radiance in the 270–500 nm range with a wide swath of 2600 km, enabling daily global coverage (Schoeberl et al., 2006). While originally designed to map ozone and its vertical distribution, OMI's hyperspectral capability also allows retrieval of trace gases such as $NO_2$, $SO_2$, HCHO, and BrO (Platt, 1994). OMI employs two aerosol

inversion schemes: OMAERO, which uses up to 19 channels between 330–500 nm for multi-wavelength aerosol optical depth retrievals, and OMAERUV, which applies two near-UV wavelengths to determine aerosol extinction optical depth and absorption properties. The OMAERUV algorithm leverages the low UV surface reflectance (except over snow) and the



enhanced sensitivity of molecular scattering to aerosol absorption in the UV (Torres et al., 1998, 2005), making retrievals feasible even over bright arid and semi-arid regions.

The OMAERUV algorithm uses reflectance measurements at 354 nm and 388 nm to derive extinction optical depth and SSA at 388 nm, as well as the UV Aerosol Index. Retrievals rely on pre-calculated reflectances for 21 aerosol microphysical models spanning three main aerosol types: dust, carbonaceous smoke, and weakly absorbing sulphate aerosols (Torres et al., 2007; Jethva and Torres, 2011). Aerosol type separation is aided by UVAI and complementary CO data from Aqua-AIRS. For a chosen aerosol type, extinction and SSA are inferred from spectral contrast and reflectance, with aerosol layer height constrained using monthly climatology from CALIOP. Large OMI pixels increase the likelihood of sub-pixel cloud contamination, leading to overestimation of extinction optical depth and underestimation of single scattering co-albedo, though such errors partly cancel in absorption optical depth (AAOD) retrievals. The OMAERUV results have been validated against airborne (Livingston et al., 2009), ground-based (Torres et al., 2007), and satellite (Ahn et al., 2008) observations, demonstrating robust performance despite known limitations.

## 2.3 MODIS fire data

The MODIS Thermal Anomalies and Fire products—MOD14 from Terra and MYD14 from Aqua—provide near-daily global fire detection at a spatial resolution of 1 km (Justice et al., 2002). Active fires are identified primarily through mid-infrared signals around 4 μm, and each fire pixel is described by its geographic location, Fire Radiative Power (FRP), and an associated confidence value. Terra records two observations per day, while Aqua adds two more, resulting in up to four daily fire detections for many regions. These observations are processed into Level-3 global composites, including daily and 8-day summaries, which are valuable for





monitoring fire activity across regional and global scales. In this work, the MOD14 dataset was used to identify forest fires in areas affected by biomass burning.

The detection method applies a contextual threshold approach: radiance at 4 µm for each pixel is compared against the background estimated from neighboring clear-sky pixels. A pixel is

classified as a fire if its brightness temperature exceeds a set threshold and shows a strong thermal contrast with the 11 µm channel. Additional filtering steps reduce false positives from effects such as sun glint, instrument noise, or desert boundaries. Fire detections are then assigned confidence levels, allowing users to apply stricter thresholds depending on application needs. The MOD14 and MYD14 datasets are distributed as Level-2 swath files in HDF format

and regridded into Level-3 products with 1 km resolution. With Collection 6.1 improvements, the products provide more accurate detections and reduced error rates compared to earlier releases. Supplementary information includes cloud and water masks, quality indicators, and classification flags that help distinguish vegetation fires from other heat sources like volcanoes or gas flares. These products have undergone extensive validation and are regarded as robust

tools for global fire monitoring.

## 2.4 Reanalysis data

The Modern-Era Retrospective analysis for Research and Applications, Version 2 (MERRA-2) (Bosilovich et al., 2015; Gelaro et al., 2017), is a global reanalysis dataset produced by NASA's Global Modeling and Assimilation Office. It provides a continuous record of atmospheric

conditions beginning in 1980 and extending to the present. MERRA-2 is generated using the Goddard Earth Observing System (GEOS) model and data assimilation system, which ingests a wide range of satellite observations along with conventional meteorological measurements. The dataset includes variables such as temperature, humidity, winds, precipitation, radiation, and aerosol properties, making it highly useful for studying both weather and climate. By



integrating observations with model outputs, MERRA-2 provides a spatially and temporally consistent picture of the Earth's atmosphere at resolutions suitable for regional to global analyses.

One of the key strengths of MERRA-2 is its inclusion of aerosol data, which improves estimates of aerosol–radiation and aerosol–cloud interactions compared to its predecessor, MERRA. It assimilates observations from instruments such as MODIS, MISR, and AVHRR to represent aerosols like dust, black carbon, organic carbon, and sea salt. These improvements enable better evaluation of air quality, radiation balance, and long-term climate trends. MERRA-2 is available at sub-daily to monthly timescales and provides output on a uniform global grid, allowing users to analyze variability ranging from diurnal cycles to multi-decadal changes. Its wide adoption across atmospheric and climate research highlights its importance as a reliable resource for applications spanning from air pollution studies to energy balance and climate change assessments.

## 3 Biomass burning regions

Biomass burning aerosols constitute a substantial portion of primary combustion aerosol emissions, with major contributions from Africa (52%), South America (15%), Equatorial Asia (10%), Boreal forests (9%), and Australia (7%) (Van der Werf et al., 2010). Three main sources of biomass burning (BB) aerosols are: (1) forest fires, (2) agricultural crop residue burning, and (3) grassland fires. The properties of BB aerosols—including their composition, size, and mixing state—play a crucial role in determining the optical characteristics of smoke plumes in the atmosphere. These optical properties are critical in influencing how BB aerosols impact the Earth's energy balance. Specifically, how these aerosols affect the radiative flux at the top of the atmosphere by scattering and absorbing incoming shortwave radiation (aerosol-radiation



interactions) and by altering cloud properties (aerosol-cloud interactions). The uncertainty in estimating the optical properties of BB aerosols is particularly significant in numerical models, since such variations in the prescribing dataset of BB aerosols in climate models may lead to unrealistic predictions. It has been reported that the characteristics of BB aerosols in climate

models carries significant uncertainty, particularly regarding their composition and optical properties (Bond et al., 2013; Andreae & Meinrat., 2019).

Amazon forest fires, with their profound ecological and atmospheric implications, have garnered significant scientific attention due to their impact on global climate and biodiversity. Fires in this region not only detrimental to its own rich biodiversity but also contribute to

substantial emissions of aerosols and greenhouse gases, altering regional and global climate systems. Consequently, extensive field research has been conducted to understand the dynamics of these fires, their aerosol emissions, and their broader environmental effects. Some of the field experiments conducted to study the aerosols over Amazon forest are: Amazonian Aerosol Characterization Experiment (AMAZE) (Martin et al., 2010; Chen et al., 2015), Large-

Scale Biosphere-Atmosphere Experiment in Amazonia (LBA) (Nobre et al., 1996; Avissar et al., 2002; Nobre et al., 2009), Aerosol Characterization Experiment (ACE-1 and ACE-2) (Bates et al., 1998; Raes et al., 2000; Quinn et al., 2005), Amazonian Atmospheric Radiation Measurement (ARM) Site (Nobre et al, 2004), and GoAmazon (Green Ocean Amazon) Experiment (Martin et al., 2016). Figure 1 shows seasonal MODIS fire counts as red dots. The

specific area of study is highlighted by the yellow box.



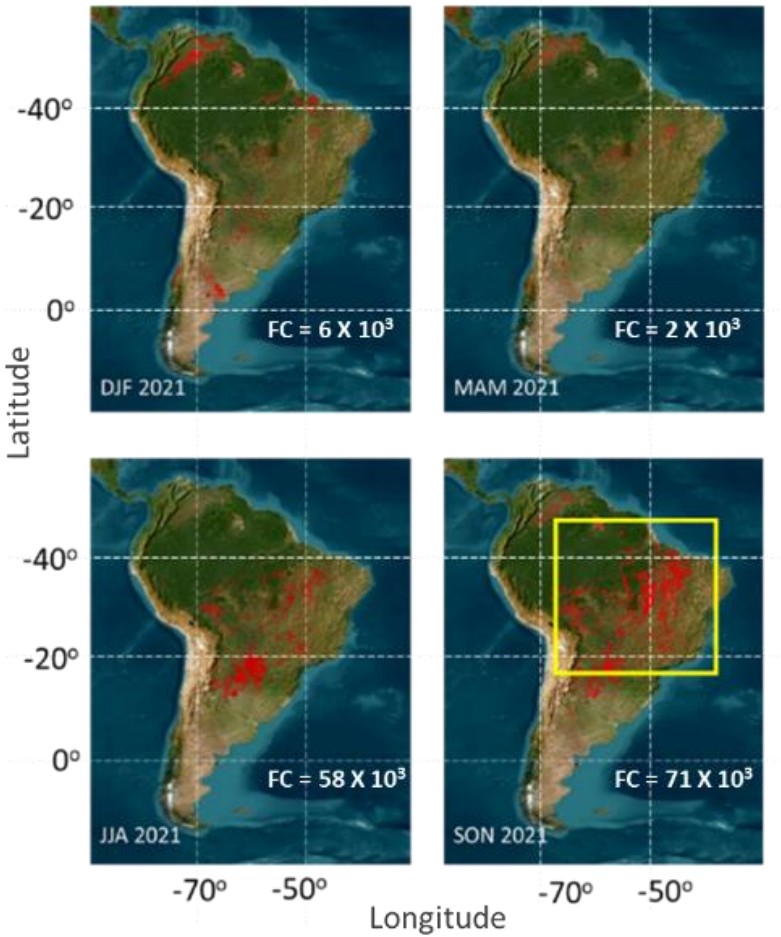

**Fig 1**. Seasonal fire counts (marked by red spots) detected by MODIS (Aqua/Terra) over South America. Corresponding fire count (FC) values are indicated in the figure. The region of interest is outlined in yellow. (Image source: NASA FIRMS; Link:

5    https://firms.modaps.eosdis.nasa.gov/)

South African forest fires is the largest contributor of biomass burning aerosols. Various field campaigns such as SAFARI 2000 (Southern Africa Regional Science Initiative 2000) (Bergstrom et al., 2003), ACTIVATE (Atlantic Circulation and Climate Experiment) (Liu et al., 2025), CAPE (Cape Peninsula Atmospheric Pollution Experiment)(Molepo et al., 2019),





FIREX (Fire Influence on Regional and Global Environments Experiment)(Azechi, 2006), and

CHAPS (CHAnnelled Aerosol Plume Study) were conducted here to understand the complex

interactions between biomass burning aerosols, cloud formation, and climate impacts,

providing critical data for climate modelling and environmental policy formulation. Figure 2

5    shows seasonal MODIS fire counts as red dots. The specific area of study is highlighted by the

yellow box.

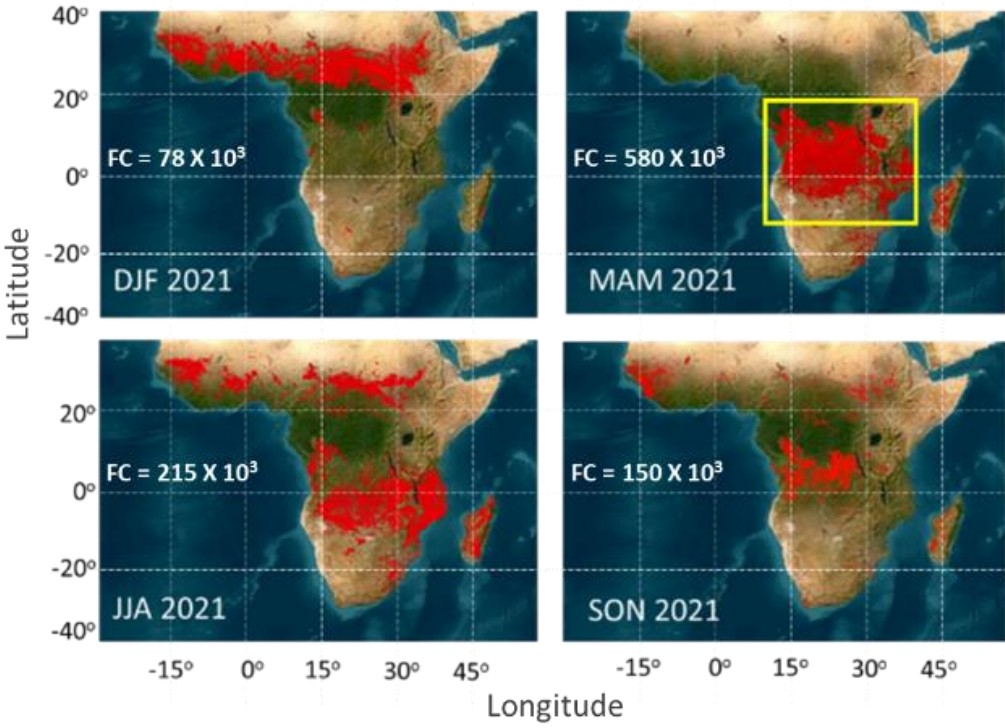

**Fig 2** Seasonal fire counts (marked by red spots) detected by MODIS (Aqua/Terra) over

South Africa. Corresponding fire count (FC) values are indicated in the figure. The region of

10    interest is outlined in yellow. (Image source: NASA FIRMS; Link:

https://firms.modaps.eosdis.nasa.gov/)





**Monthly variation in fire count:** Figure 3 presents the average monthly fire count (in thousands) over the Amazon forest from 2014 to 2022, depicted by the red line, along with the standard deviation shown as a shaded area. The fire count reaches it maximum during August and September during the dry season, associated with its well-defined seasonal variability. These months are characterized by lower humidity and less rainfall creating conditions conducive to fires, often exacerbated by human activities such as deforestation and agricultural burning. The standard deviation is relatively narrow during the low fire count months but widens significantly during the peak fire season, indicating greater variability in fire occurrences during this period.

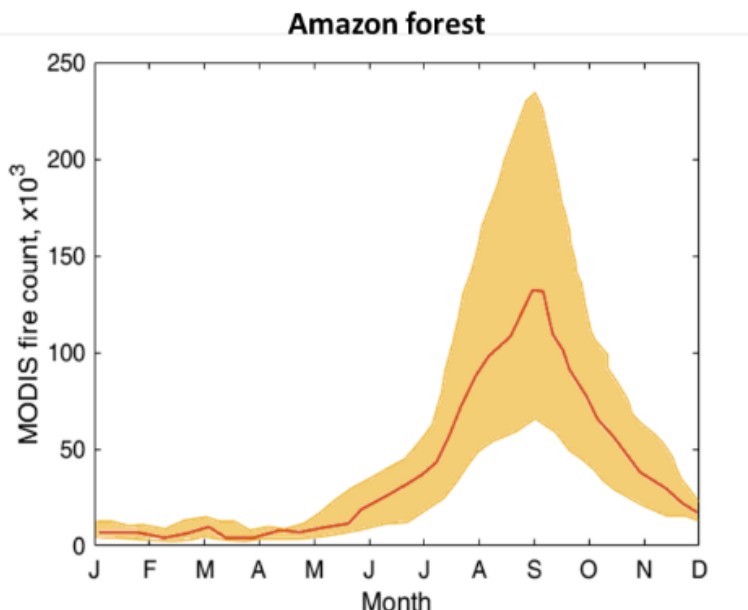

**Fig 3.** Average (red line) and standard deviation of the fire count over amazon forest for the years 2014 – 2022




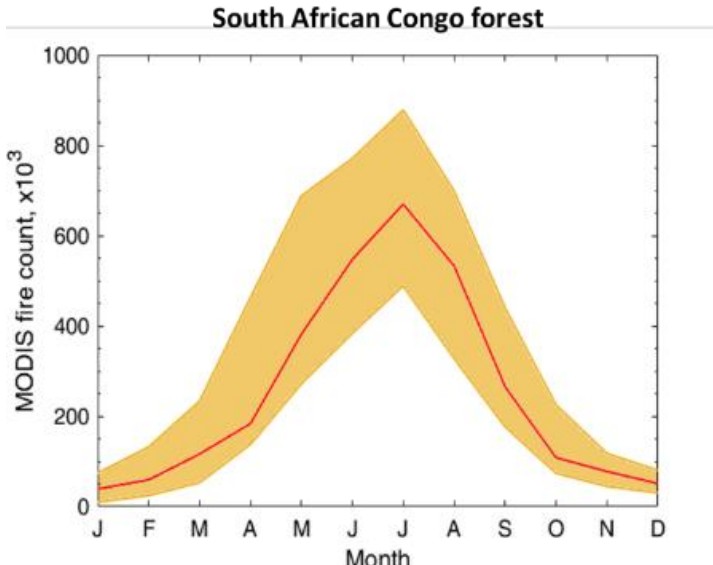

**Fig 4.** Average (red line) and standard deviation of the fire count over South African forest for the years 2014 – 2022

Figure 4 illustrates the average monthly fire count (in thousands) over the South African Congo

forest from 2014 to 2022. Similar to the Amazon, there is distinct and seasonal trend in fire

activity, with the highest fire counts observed during the dry season, peaking around June to

August. The peak fire count in the South African Congo forest is substantially higher than in

the Amazon, reaching nearly 800,000 in the month of August. The standard deviation follows

a similar pattern, being narrow during months with low fire activity and expanding during the

peak months, indicating higher variability in fire occurrences during the dry season. This

region's fire activity is influenced by both natural climatic conditions and human-induced

factors such as land clearing for agriculture.

Comparison of the two regions (Fig 3 and 4), both the Amazon and South African Congo

forests, exhibit clear seasonal trends in fire activity, with peaks during their respective dry

seasons. However, the magnitude of fire counts differs significantly between the two regions.





The South African Congo forest experiences a much higher peak fire count, with a maximum of around 800,000 in August, compared to the Amazon's peak of nearly 200,000 in September. Additionally, the South African Congo forest shows a more symmetrical fire count pattern around its peak, while the Amazon forest's fire activity rises sharply and declines more gradually. The variability in fire counts, as indicated by the standard deviation, is higher in the South African Congo forest, suggesting greater fluctuations in fire occurrences during the peak season. These differences highlight the varying environmental conditions and human impacts on fire activity in these two critical forest regions.

**Variation of SSA with MODIS fire count:**

**a) Amazon forest:**

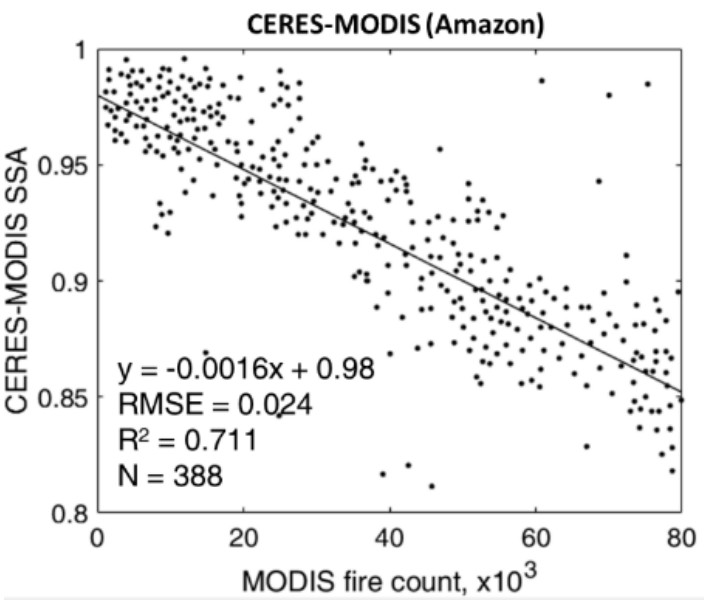

**Fig 5** Scatter plot of fire count vs CERES-MODIS SSA over Amazon forest





Figure 5 illustrates the scatter plot of fire count versus CERES-MODIS SSA over the Amazon forest, using 388 such datasets, providing a robust sample for this analysis. The data shows a clear negative correlation between SSA and fire count, as indicated by the regression line with a slope of -0.0016. This negative slope suggests that as the fire count increases, the SSA

5    decreases, which is consistent with the expectation that higher fire counts lead to an increase in absorbing aerosols like black carbon, thereby lowering the SSA. The coefficient of determination ($R^2$) is 0.711, indicating that approximately 71.1% of the variability in SSA can be explained by the fire count. The RMSE of 0.024 shows the average difference between the observed SSA values and those predicted by the regression model. Overall, the CERES-

10   MODIS data supports the hypothesis that increased fire activity leads to lower SSA due to the presence of more absorbing aerosols.

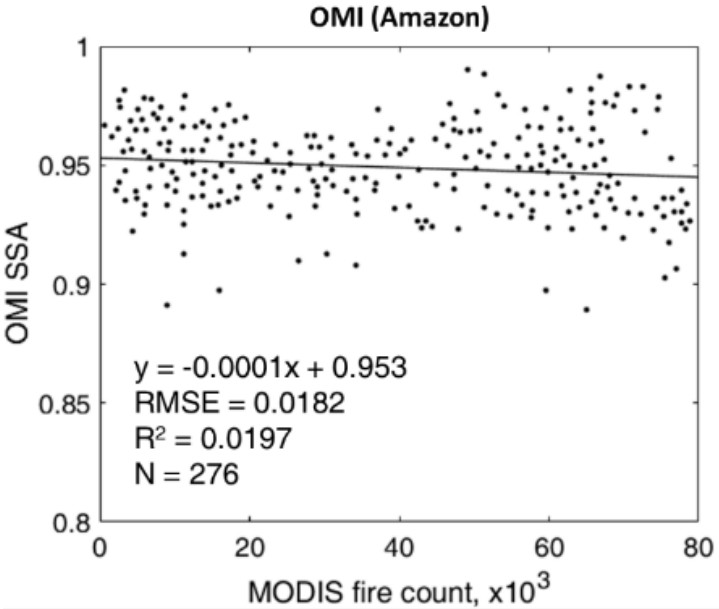

**Fig 6** Scatter plot of fire count vs OMI SSA over Amazon forest





Figure 6 presents the scatter plot of fire count versus OMI SSA over the Amazon forest using 276 datasets. Unlike the CERES-MODIS data, the OMI SSA shows a much weaker relationship with fire count. The regression line has a very slight negative slope of -0.0001, indicating a negligible decrease in SSA with increasing fire count. The coefficient of determination ($R^2$) is

only 0.0197, suggesting that fire count explains less than 2% of the variability in SSA. The RMSE is 0.0182, slightly lower than that of the CERES-MODIS data, but this does not significantly affect the overall weak correlation. The OMI data suggests that fire count has a minimal impact on SSA, which could be due to various factors, including differences in the retrieval algorithms or the spectral ranges used by OMI.

Comparing the two figures (5 and 6), it is evident that CERES-MODIS SSA shows a stronger negative correlation with fire count than OMI SSA. The CERES-MODIS data indicates a clear decrease in SSA with increasing fire counts, supporting the hypothesis that higher fire activity leads to more absorbing aerosols and thus lower SSA. In contrast, the OMI data shows a very weak and almost negligible correlation between SSA and fire count. Overall, while CERES-

MODIS data aligns with the anticipated relationship between SSA and fire count, the OMI data does not show a significant correlation, suggesting the need for further investigation into the factors influencing these measurements.

**b)  South African forest**



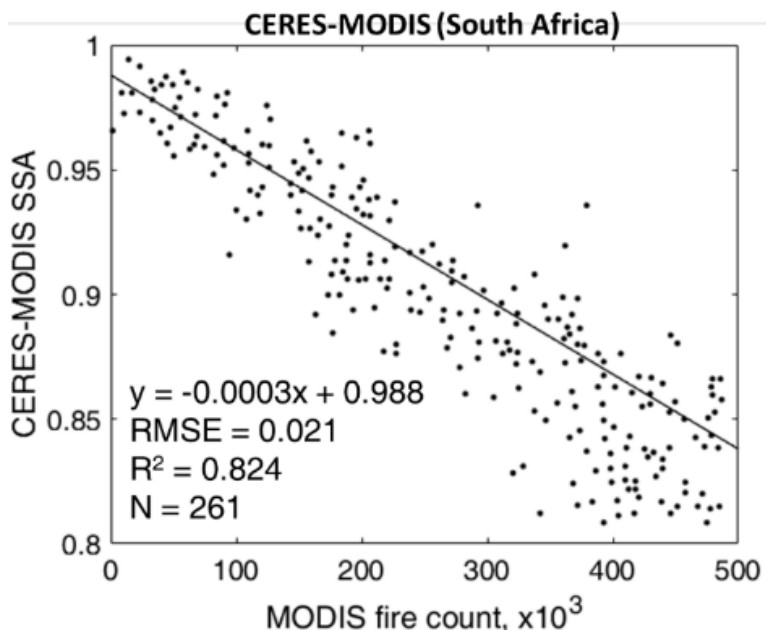

**Fig 7** Scatter plot of fire count vs CERES-MODIS SSA over South African forest

Figure 7 shows a scatter plot of fire count versus CERES-MODIS SSA over the South African

5    forest. The plot with a sample size (N) of 261 reveals a strong negative correlation between

SSA and fire count, as evidenced by the regression line with a slope of -0.0003, implying higher

absorption by aerosols due to increased fire activity. The coefficient of determination ($R^2$) is

0.824, suggesting that 82.4% of the variability in SSA can be explained by the fire count. The

RMSE is 0.021, indicating a good fit of the regression model to the observed data. Results

10    indicate that CERES-MODIS data are capable of capturing the relation between increased fire

counts and the corresponding decrease in SSA due to the presence of more absorbing aerosols

in the South African forest.





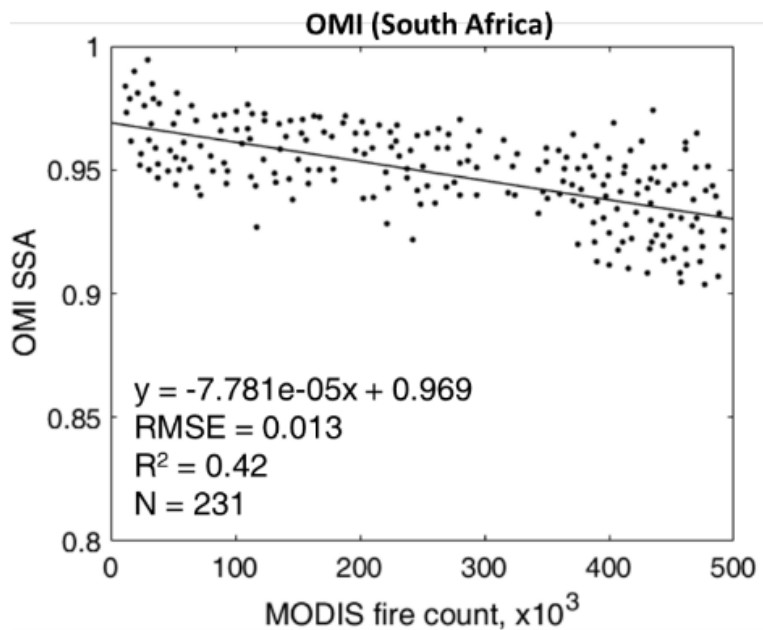

**Fig 8** Scatter plot of fire count vs OMI SSA over South African forest

Figure 8 depicts the scatter plot of fire count versus OMI SSA over the South African forest. The plot with a sample size 231, shows a weaker negative correlation between SSA and fire count, with the regression line having a slope of -7.781e-05. The coefficient of determination ($R^2$) is 0.42, indicating that only 42% of the variability in SSA can be explained by the fire count, which is significantly lower than that observed with CERES-MODIS. The RMSE is 0.013, suggesting a reasonable fit of the regression model but highlighting some discrepancies. The weaker relationship between SSA and fire count in the OMI data highlights its inability to effectively capture the presence of absorbing aerosols over biomass burning regions.

Despite the higher fire count in South Africa compared to the Amazon, the reduction in SSA with increasing fire count is less steep. This difference can be attributed to the distinct fire season characteristics and vegetation types in each region. In the Amazon, fires occur in a shorter but more intense peak around August–September, leading to a sharp increase in



absorbing aerosols such as black carbon over a brief period (as seen before in Fig 3 and 4), which strongly reduces SSA. Additionally, the Amazon consists of dense forests, where biomass burning releases a higher fraction of black carbon and organic aerosols, further lowering SSA. In contrast, South Africa experiences a much broader fire season, extending

from June to September, with fire activity spread over a longer duration. The region is also dominated by savanna and grassland fires, which generally produce a different aerosol composition, with a relatively higher fraction of scattering aerosols. This difference in vegetation and burning patterns leads to a weaker reduction in SSA in South Africa compared to the Amazon.

Comparing figures (7 and 8) with (Figs 5 and 6), it is evident that the relationship between SSA and fire count varies between the Amazon and South African forests. In both regions, CERES-MODIS shows a strong negative correlation between SSA and fire count, with high $R^2$ values (0.711 for the Amazon and 0.824 for South Africa), indicating that fire activity significantly impacts SSA. However, the slope of the regression line is steeper in the Amazon (-0.0016)

compared to South Africa (-0.0003), suggesting a more pronounced decrease in SSA with increasing fire counts in the Amazon. In contrast, OMI data for both regions show weaker correlations, with $R^2$ values of 0.0197 for the Amazon and 0.42 for South Africa. The slope of the regression line for OMI is also much smaller, indicating a minimal decrease in SSA with increasing fire counts. This indicates inability of OMI observations to effectively capture the

SSA variations with biomass burning, and fire activities. To obtain a deeper understanding on the applicability of these observations across various seasons, in the following section we explore the seasonal variation of AOD and SSA.





**Seasonal variation in AOD and SSA:**

AOD and SSA of various aerosol types across different seasons has been studied, specifically

for the Amazon forest, and South African forest, using the CERES-MODIS algorithm.

**Amazon forest:**

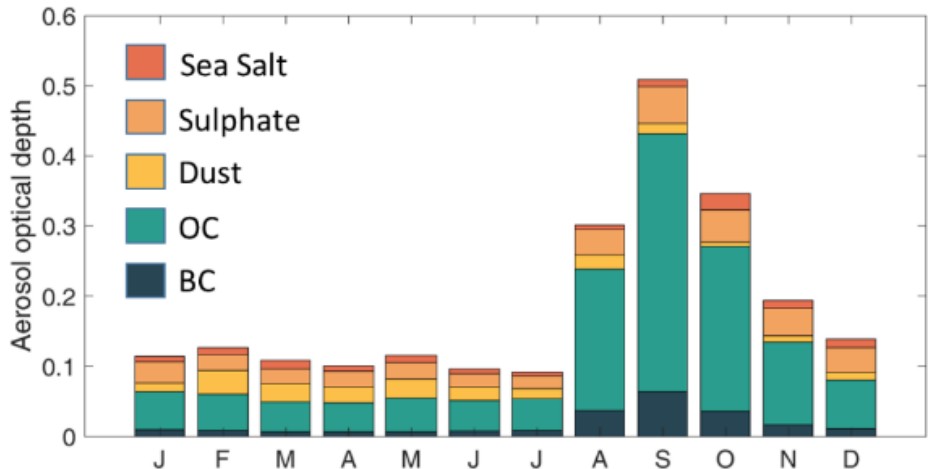

**Fig 9.** Variations in monthly mean optical depth of various aerosol types over amazon forest

Figure 9 illustrates the variations in the monthly mean AOD for various aerosol types over the

Amazon forest. Figure shows that during August to September, the AOD reaches it maximum,

10 and Organic Carbon (OC) and Black Carbon (BC) are the predominant aerosol species. This

increase is consistent with the higher fire activity observed during these months (as discussed

in Fig 5.10), as biomass burning releases large amounts of carbonaceous aerosols into the

atmosphere. Other aerosol types, such as sulphates, dust, and sea salt, show relatively minor

contributions throughout the year but exhibit slight increases during the peak fire months,

15 possibly due to secondary interactions with the primary fire emissions. In addition to this, the

organic and black carbon combinedly contributes to nearly half of the AOD during all the



seasons, except during the peak months, where its contribution to total AOD is more than 60-70 %. These observations are consistent with the SSA variability captured by the CERES-MODIS algorithm.

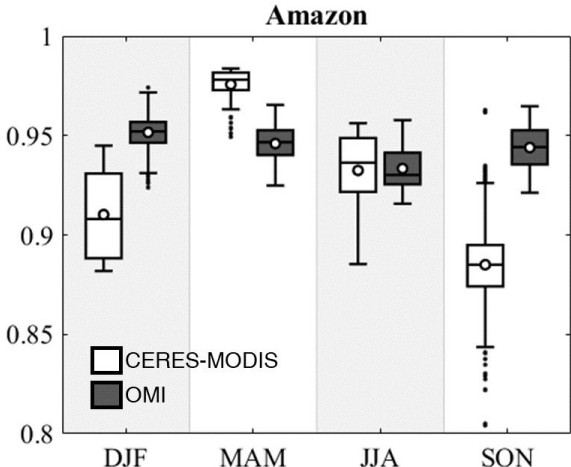

**Fig 10** Boxplots indicating seasonal SSA over the Amazon forest. The white boxes represent SSA values retrieved by CERES-MODIS, and the black boxes represent SSA values retrieved by OMI. Each boxplot displays the median (central line), the interquartile range (IQR) from the 25th percentile (bottom edge of the box) to the 75th percentile (top edge of the box), and the whiskers, which extend to 1.5 times the IQR or to the maximum and minimum values within this range. Outliers beyond the whiskers are plotted as individual points.

Figure 10 presents boxplots indicating the seasonal SSA values over the Amazon forest for both CERES-MODIS and OMI. The white boxes represent CERES-MODIS SSA values, while the black boxes represent OMI SSA values. The figure highlights a clear seasonal variation in SSA, with lower SSA values observed during the dry season (June to November), particularly in August and September. During the wet season (December to May), SSA values are higher and more stable, reflecting lower fire activity and a reduced concentration of absorbing





aerosols. The interquartile range (IQR) and whiskers suggest that CERES-MODIS data has greater variability during the dry season, while OMI data shows more consistent SSA values across all seasons. Further, it can be noticed that, the CERES-MODIS data shows more pronounced seasonal variations compared to OMI, which tends to have more stable SSA values across seasons. This difference is due to the higher sensitivity of CERES-MODIS to absorbing aerosols during periods of intense biomass burning.

In summary, the contribution of OC and BC due to increased fire activities during the dry seasons results in high AOD, and low SSA, which are well-captured by the CERES-MODIS scheme, while the OMI observations do not capture these variations effectively.

**South African forest**:

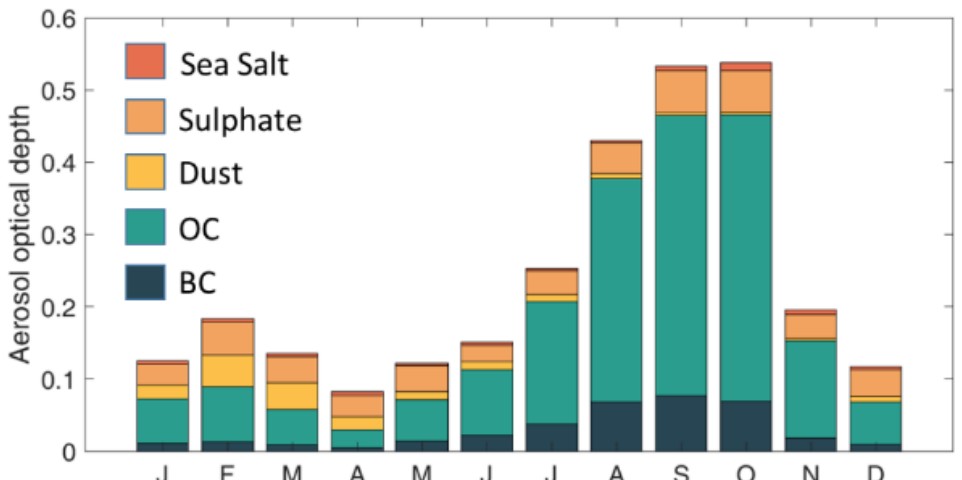

**Fig 11** Variations in monthly mean optical depth of various aerosol types over South African forest



Figure 11 illustrates the variations in the monthly mean AOD for various aerosol types over the South African forest. The data reveals significant seasonal fluctuations, with notable increases in AOD during the dry season from July to October. During this period, Organic Carbon and Black Carbon are the dominant contributors to the total AOD, reflecting the influence of

biomass burning. The peak in aerosol loading occurs in August and September, where AOD values are highest, indicating intense fire activity. Sulphates, dust, and sea salt also show increased levels during these months but contribute less significantly to the overall AOD compared to OC and BC. The pattern suggests that biomass burning is the primary source of aerosols during the dry season in the South African forest. Despite having similar variability in

the case of maximum AOD's, the South African forest fires are captured over a more prolonged duration, indicated by the increased number of the peak months.

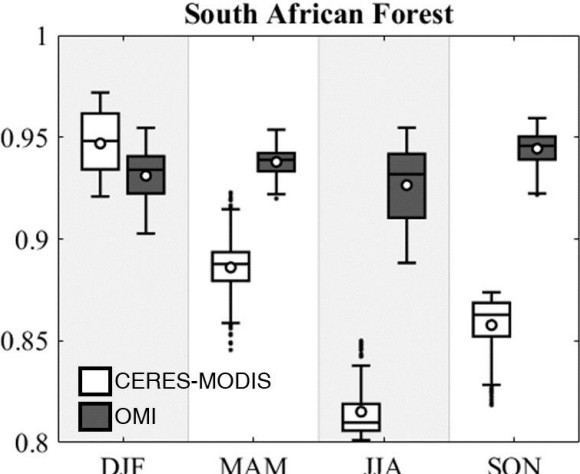

**Fig 12.** Boxplots indicating seasonal SSA over the South African forest. The white boxes

represent SSA values retrieved by CERES-MODIS, and the black boxes represent SSA values retrieved by OMI. Each boxplot displays the median (central line), the interquartile range (IQR) from the 25th percentile (bottom edge of the box) to the 75th percentile (top edge of the box),





and the whiskers, which extend to 1.5 times the IQR or to the maximum and minimum values within this range. Outliers beyond the whiskers are plotted as individual points

Figure 12 presents boxplots indicating the seasonal SSA values over the South African forest, comparing data from CERES-MODIS (white boxes) and OMI (black boxes). CERES-MODIS shows distinct seasonal variations in SSA, with lower values observed during the dry season (June to November), particularly in July and August. The decrease in SSA during these months corresponds to the increase in absorbing aerosols such as BC and OC due to biomass burning, as seen in Fig 11. The CERES-MODIS data shows more pronounced seasonal variations with a significant drop in SSA during the dry season, while the OMI data exhibits less pronounced but still noticeable seasonal trends. The boxplots indicate that CERES-MODIS is more sensitive to changes in aerosol composition and loading, resulting in greater variability in SSA values compared to OMI.

## 4 Oceanic regions

### 4.1 Relatively clean ocean

The South Pacific Ocean is an area largely isolated from significant continental pollution sources, making it an ideal location for studying relatively clean oceanic regions. The vast expanse of the South Pacific, far from major landmasses, ensures that it experiences low levels of anthropogenic aerosol input from the land. Consequently, this region provides a pristine environment for examining natural aerosol processes and their interactions with the atmosphere



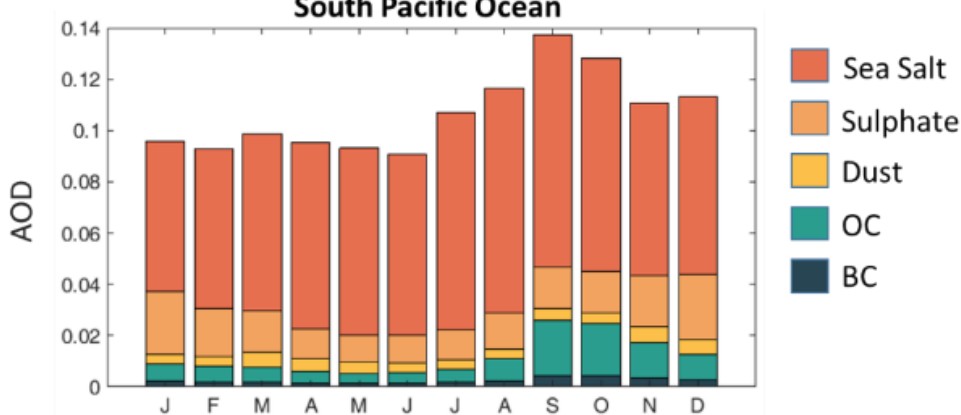

**Fig 13** Variations in monthly mean optical depth of various aerosol types over South Pacific Ocean.

Figure 13 shows the monthly variations in AOD over the South Pacific Ocean, highlighting the contributions from different aerosol types: Sea Salt, Sulphate, Dust, Organic Carbon, and Black Carbon. Sea Salt aerosols dominate throughout the year, as expected in an oceanic environment. The AOD remains relatively stable from January to July, with slight increase in Sulphate and Dust during this period. However, from August to October, there is a noticeable

peak in AOD, indicating higher aerosol concentrations. This peak is primarily driven by an increase in Sea Salt and OC, with smaller contributions from Sulphate and BC. This could be attributed to seasonal atmospheric conditions, such as stronger winds and changes in oceanic activity, which enhances emission and transport of aerosols. Overall, the South Pacific Ocean remains relatively clean, with natural aerosols like sea salt being the dominant contributors,

while seasonal variations influence aerosol loading in this remote region.



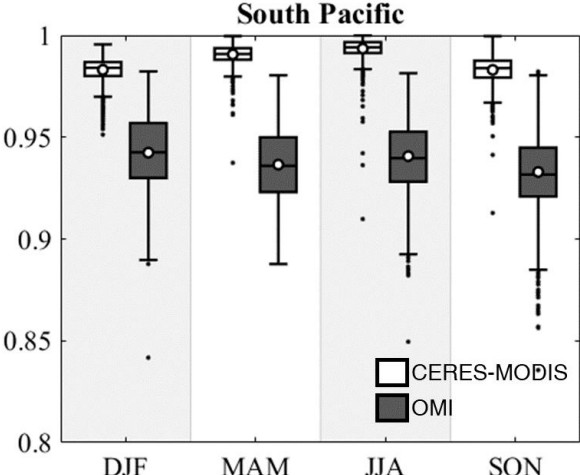

**Fig 14.** Boxplots indicating seasonal SSA over the South Pacific Ocean. The white boxes represent SSA values retrieved by CERES-MODIS, and the black boxes represent SSA values retrieved by OMI. Each boxplot displays the median (central line), the interquartile range (IQR) from the 25th percentile (bottom edge of the box) to the 75th percentile (top edge of the box), and the whiskers, which extend to 1.5 times the IQR or to the maximum and minimum values within this range. Outliers beyond the whiskers are plotted as individual points.

Figure 14 presents boxplots of the seasonal SSA over the South Pacific Ocean, comparing values retrieved by CERES-MODIS (white boxes) and OMI (black boxes). The data reveals that SSA values are generally high throughout the year, indicative of the predominance of scattering aerosols, such as Sea Salt, which are less absorbing. Both CERES-MODIS and OMI data show slight seasonal variations in SSA, with the highest values occurring during the austral summer (DJF) and autumn (MAM), and slightly lower values during winter (JJA) and spring (SON). The interquartile ranges (IQR) and whiskers suggest that OMI data exhibits more variability than CERES-MODIS, with a wider spread of SSA values and more outliers, particularly in JJA and SON. However, this high variability in SSA values observed in the OMI





data throughout the year suggests potential retrieval errors or inconsistencies. This variability contrasts with the more stable SSA values recorded by CERES-MODIS, which align better with the expected aerosol characteristics of this pristine environment.

Analysis of the boxplot figure 14 alongside the previously discussed bar graph of monthly mean AOD (fig 13), reveals a coherent picture regarding the aerosol characteristics in the South Pacific Ocean. The high SSA values observed in CERES-MODIS data throughout the year are consistent with the dominance of scattering aerosols like sea salt over this region. The slight decrease in SSA during spring (SON) aligns with the observed peaks in AOD during these months. These peaks indicate an increased presence of aerosols, which slightly lowers SSA due to the higher proportion of absorbing aerosols. The box plot highlights the superior performance of CERES-MODIS over OMI in retrieving SSA over the South Pacific (a relatively clean ocean). CERES-MODIS consistently captures higher SSA values, consistent with the dominance of scattering aerosols. While CERES-MODIS also captures a slight increase in SSA during spring, OMI consistently underestimates SSA, with values ranging between 0.86 and 0.96, averaging around 0.94, throughout the year suggesting potential errors in its retrieval algorithm.

## 4.2 Polluted Ocean

The Arabian Sea and Bay of Bengal are ideal regions for studying polluted oceans due to the significant continental outflow of pollution from nearby landmasses. These areas receive a substantial amount of absorbing aerosols, such as black carbon and organic carbon, from industrial and vehicular emissions in the surrounding land regions. During periods of heavy winds, sea salt aerosols are also produced, adding to the aerosol load. Additionally, dust from the Thar Desert and nearby Arabian regions further contributes to the aerosol mix, making them rich in both natural and anthropogenic aerosols.





Figure 15 illustrates the monthly variations in AOD for different aerosol types over the Arabian Sea. The data shows significant seasonal fluctuations, with the highest AOD values occurring during June - August. This period coincides with the southwestern monsoon season or the Indian summer monsoon, characterized by heavy winds that increase the production of sea salt

5    aerosols, the dominant component during these months. Dust also plays a significant role, particularly from the Thar Desert and Arabian regions, contributing to the high AOD. Sulphates, organic carbon and black carbon are present throughout the year, with increased levels in the summer months, reflecting the continental outflow of pollution from surrounding industrial and urban areas.

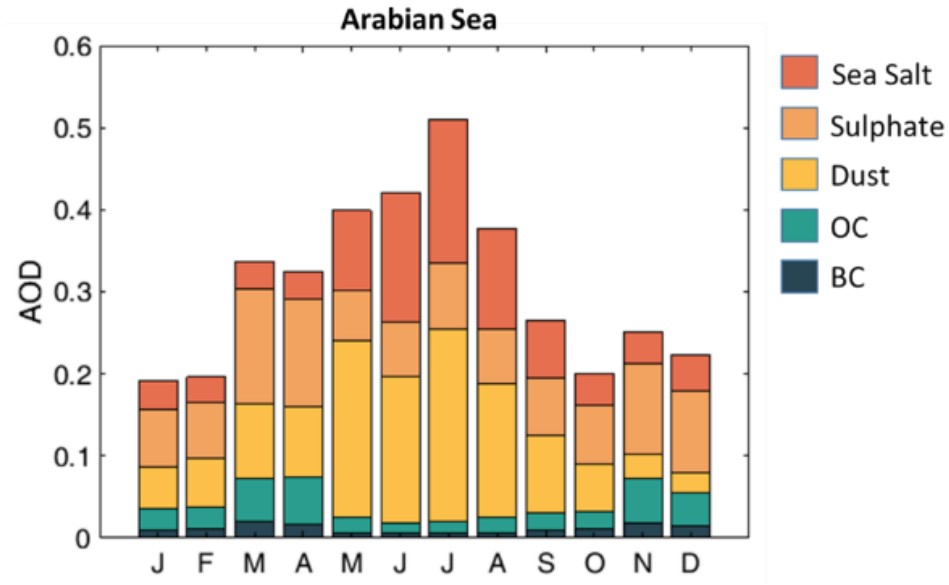

**Fig 15** Monthly variation of aerosol type AOD over Arabian Sea




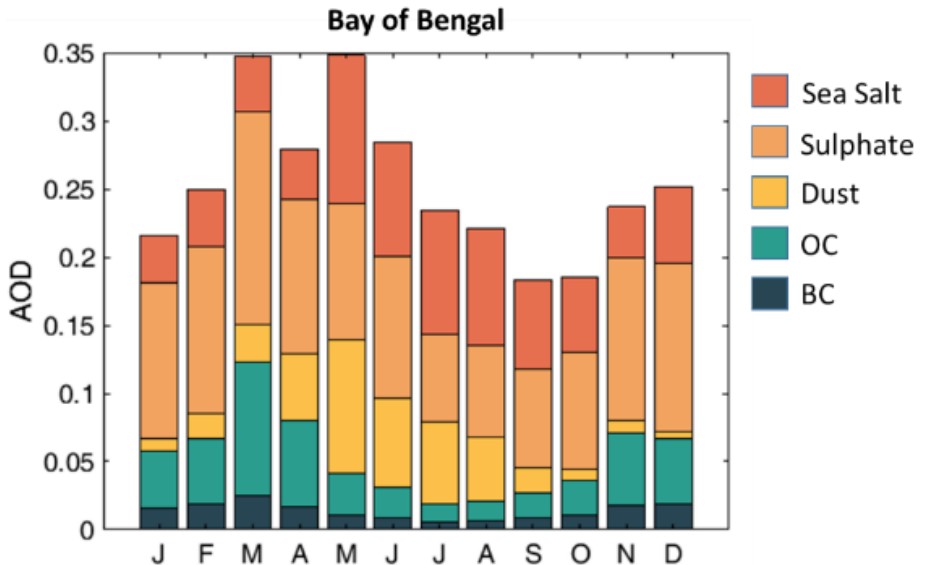

**Fig 16** Monthly variation of aerosol type AOD over Bay of Bengal

Figure 16 shows the monthly variations in AOD for various aerosol types over the Bay of

Bengal. The aerosol composition in this region is primarily influenced by nearby cities like

Chennai, Vishakhapatnam, and Bhuvaneswar. Highest AOD is observed during MAM, with

peaks driven mainly by sulphates and sea salt, reflecting the combined influence of continental

pollution and marine aerosols. Dust, OC, and BC also contribute significantly, especially

during the pre-monsoon period, indicating the transport of aerosols from nearby landmasses

and industrial activities. During the winter months (December to February), there is an increase

in BC and OC concentrations, primarily due to aerosols being transported from the Indo-

Gangetic Plain and eastern India, where biomass burning and industrial emissions are

prevalent. The monsoon season (June to September) brings a significant reduction in BC and

OC levels, as strong winds and heavy rainfall help cleanse the atmosphere. In the post-monsoon

period (October and November), aerosol concentrations gradually rise, as atmospheric





conditions stabilize and allow for the accumulation of both natural and anthropogenic aerosols.

(Satheesh et al., 2001; Satheesh 2002; Moorthy et al., 2003; Satheesh et al., 2006; Babu et al., 2012)

Comparing the two figures (15 and 16), both the Arabian Sea and Bay of Bengal exhibit

significant seasonal variations in AOD. The Arabian Sea shows higher overall AOD values, particularly during the monsoon season, with sea salt and dust being the dominant components. In contrast, the Bay of Bengal experiences its highest AOD values in the pre-monsoon period, with sulphates and sea salt contributing most significantly. Both regions show substantial contributions from OC and BC, reflecting the impact of continental outflow and industrial

emissions.

**Seasonal variation in SSA:**

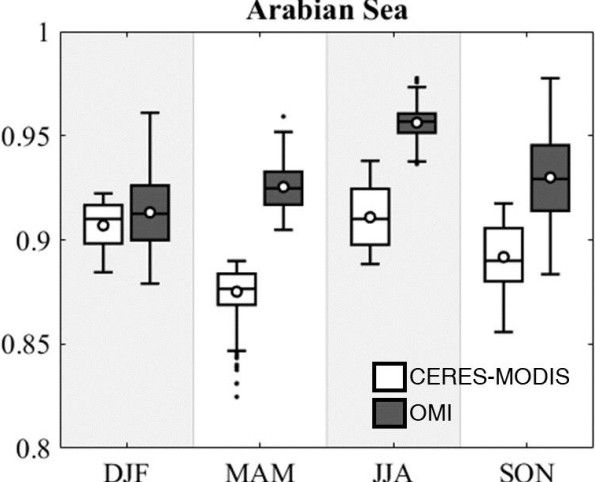

**Fig 17.** Boxplots indicating seasonal SSA over the Arabian Sea. The white boxes represent SSA values retrieved by CERES-MODIS, and the black boxes represent SSA values retrieved by

OMI. Each boxplot displays the median (central line), the interquartile range (IQR) from the 25th percentile (bottom edge of the box) to the 75th percentile (top edge of the box), and the



whiskers, which extend to 1.5 times the IQR or to the maximum and minimum values within this range. Outliers beyond the whiskers are plotted as individual points.

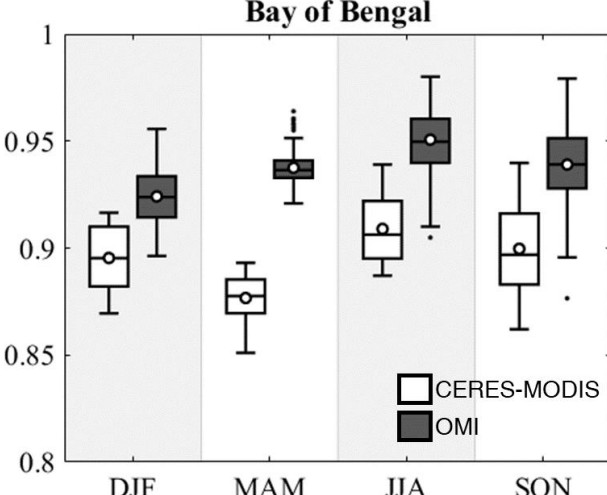

**Fig 18.** Boxplots indicating seasonal SSA over Bay of Bengal. The white boxes represent SSA values retrieved by CERES-MODIS, and the black boxes represent SSA values retrieved by OMI. Each boxplot displays the median (central line), the IQR from the 25th percentile (bottom edge of the box) to the 75th percentile (top edge of the box), and the whiskers, which extend to 1.5 times the IQR or to the maximum and minimum values within this range. Outliers beyond

the whiskers are plotted as individual points.

Figure 17 presents boxplots indicating the seasonal SSA over the Arabian Sea, comparing values retrieved by CERES-MODIS (white boxes) and OMI (black boxes). The data reveals clear seasonal variations in SSA, with the lowest values observed during the pre-monsoon season and higher values during the monsoon period. The decrease in SSA during the pre-

15 monsoon season can be attributed to the increased presence of absorbing aerosols such as black carbon and dust, which are transported over the Arabian Sea from surrounding landmasses. The



boxplots indicate that CERES-MODIS data shows more variability compared to OMI, suggesting that CERES-MODIS is more sensitive to changes in aerosol composition and loading. The interquartile ranges (IQR) and whiskers highlight the spread and distribution of SSA values, with outliers indicating extreme values likely influenced by episodic pollution events or retrieval outliers.

Figure 18 shows boxplots of seasonal SSA over the Bay of Bengal. Similar to the Arabian Sea, the Bay of Bengal also exhibits seasonal variations in SSA, with the lowest values occurring during the pre-monsoon (MAM). These lower SSA values are indicative of higher concentrations of absorbing aerosols, which are likely due to continental outflow and biomass burning activities in the surrounding regions. The CERES-MODIS data again captures more pronounced variability compared to OMI, particularly during the pre-monsoon season. The boxplots reveal that SSA values are generally higher during the post-monsoon (SON) and winter (DJF) seasons, reflecting a cleaner atmospheric environment with fewer absorbing aerosols. CERES-MODIS consistently shows greater sensitivity and variability in SSA values compared to OMI, suggesting that CERES-MODIS may be more effective in capturing the dynamic changes in aerosol composition and loading in these regions.

# 5 Land regions

## 5.1 Relatively clean land

North America and European regions are particularly well-suited for studying clean environments due to their generally lower levels of aerosol loading compared to other parts of the world. These regions benefit from stringent air quality regulations and relatively lower emissions from industrial and agricultural sources. Consequently, the cleaner atmospheric conditions in these areas provide a clearer baseline for studying natural aerosol processes. This



relative lack of anthropogenic aerosol interference allows for more accurate assessments of background aerosol levels and their interactions with atmospheric components.

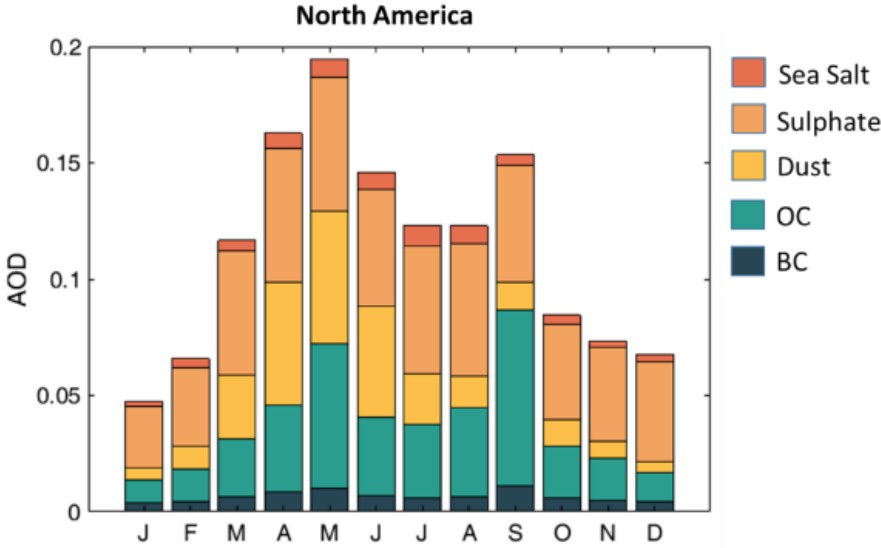

**Fig 19** Monthly variation of aerosol type AOD over North America

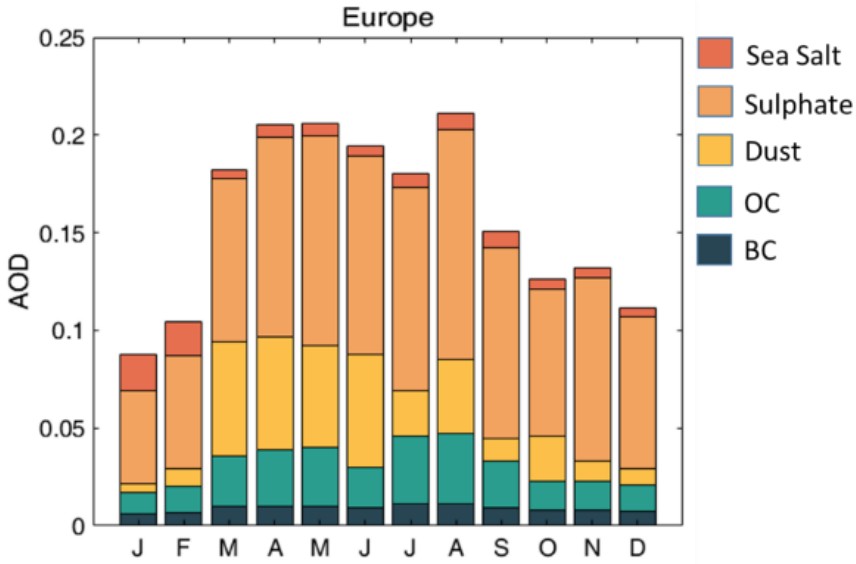



**Fig 20** Monthly variation of aerosol type AOD over Europe

Figure 19 illustrates the monthly variations in AOD for different aerosol types over North America. The data reveals that sulphates and organic carbon are the dominant contributors to

AOD throughout the year, with notable peaks from April to August. This period corresponds to the warmer months when increased industrial activity and vehicular emissions contribute to higher levels of sulphates and OC. During the months of MAM, North America experiences an increase in AOD due to the enhanced transport of dust. This seasonal rise is driven by stronger winds and drier conditions in the Sahara Desert, leading to increased dust emissions that are

carried across the Atlantic Ocean by prevailing winds (Goudie et al., 2001; Middleton and Goudie., 2001; Tanré et al., 2003; Prospero 2007; Prospero et al., 2014; Prospero et al., 2021). In addition to Saharan dust, North American deserts such as the Mojave, Sonoran, and Chihuahuan deserts also contribute to dust levels during this period. The atmospheric conditions during MAM favour the suspension and movement of these dust particles,

influencing air quality, visibility, and climate patterns in the affected regions. Influence of sea salt and black carbon aerosols are less compared to other aerosol types but show slight increases during the same period, reflecting seasonal changes in atmospheric conditions and human activities.

From March to September, OC and BC concentrations in North America increase due to

various factors, particularly wildfires. California's wildfire season, which peaks from July to September, is a significant contributor, releasing large amounts of OC and BC into the atmosphere and impacting air quality across the western U.S (Singh et al., 2010; Liang et al., 2020; Eck et al., 2023). Additionally, boreal forest fires in Canada and Alaska contribute during the June to August (JJA) period. These fires, occurring in areas with dense forest cover, emit



substantial amounts of OC and BC, with the smoke often traveling southward and affecting air quality in North America (Miller et al., 2011; Jolleys et al., 2015). The combined effects of California and boreal forest fires lead to a marked increase in OC and BC concentrations during these months, influencing atmospheric composition and air quality across the continent.

Figure 20 shows the monthly variations in AOD for various aerosol types over Europe. Similar to North America, sulphates and OC are the primary contributors to AOD, with the highest values observed from April to September. This seasonal pattern aligns with increased industrial emissions and biomass burning activities during the warmer months. During the summer months, dust AOD over Europe increases primarily due to the transport of dust from North

African deserts, especially the Sahara and the Sahel region (Gaudie et al., 2001; Stuut et al., 2009; Karanasiou et al., 2012; Marinou et al., 2017; Wang et al., 2020). Stronger winds and drier conditions in these deserts during summer lead to higher dust emissions, which are then carried across the Mediterranean by prevailing winds. Sea salt aerosols remain relatively constant throughout the year but show minor peaks in the winter months, due to increased wind

speeds over the ocean. Black carbon, while a smaller component, shows consistent levels year-round with slight increases during the winter months, reflecting residential heating and transportation emissions.

Comparing the two figures (19 and 20), both North America and Europe exhibit similar seasonal trends in AOD, with sulphates and OC being the dominant aerosol types contributing

to higher AOD values during the warmer months. However, Europe shows generally higher AOD values compared to North America, with major contribution from sulphates, which can be attributed to differences in industrial activities, energy production, and transportation emissions (Mao et al., 2014; Pozzer et al., 2015; Zhang et al., 2024). Dust aerosols are significant in both regions, but North America sees a more pronounced peak in the spring, while

Europe's dust levels peak in the summer, influenced by Sahara dust transport. Sea salt





contributions are relatively low in both regions, with slight seasonal variations. Black carbon

levels are comparable in both regions, with minor increases during colder months due to

residential heating.

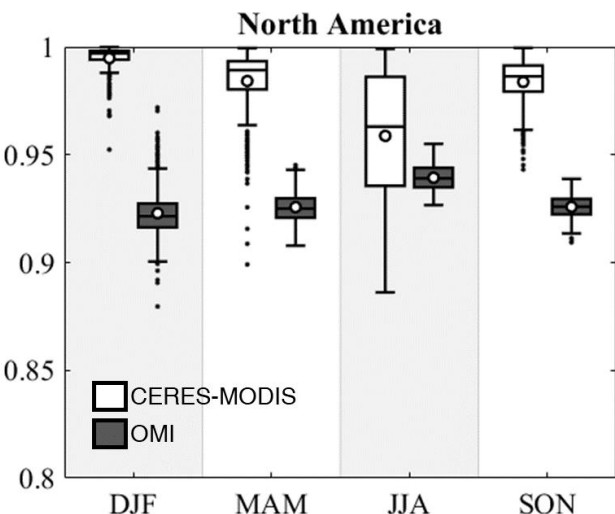

**Fig 21.** Boxplots indicating seasonal SSA over North America. The white boxes represent SSA

values retrieved by CERES-MODIS, and the black boxes represent SSA values retrieved by

10    OMI. Each boxplot displays the median (central line), the IQR from the 25th percentile (bottom

edge of the box) to the 75th percentile (top edge of the box), and the whiskers, which extend

to 1.5 times the IQR or to the maximum and minimum values within this range. Outliers beyond

the whiskers are plotted as individual points.



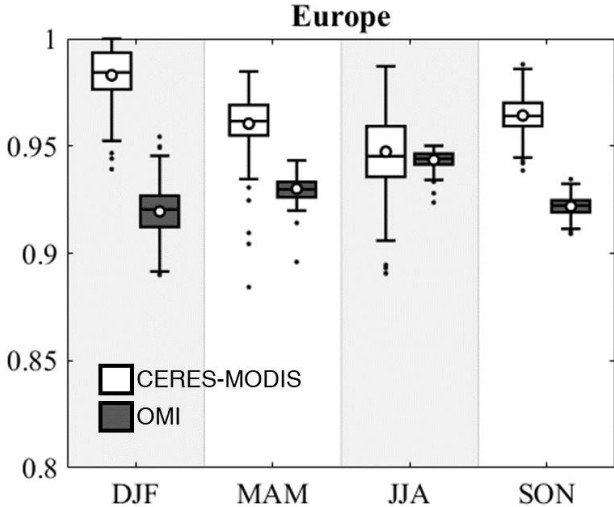

**Fig 22.** Boxplots indicating seasonal SSA over Europe. The white boxes represent SSA values retrieved by CERES-MODIS, and the black boxes represent SSA values retrieved by OMI. Each boxplot displays the median (central line), the IQR from the 25th percentile (bottom edge of the box) to the 75th percentile (top edge of the box), and the whiskers, which extend to 1.5 times the IQR or to the maximum and minimum values within this range. Outliers beyond the whiskers are plotted as individual points.

Figure 21 presents boxplots showing the SSA over North America, comparing values retrieved by CERES-MODIS (white boxes) and OMI (black boxes). The wide range of SSA values observed from March to September is primarily due to the presence of transported desert dust and absorbing aerosols from forest fires in California and the Canadian boreal forests. The boxplots indicate that CERES-MODIS data exhibits greater variability compared to OMI, particularly in the summer, suggesting that CERES-MODIS is more sensitive to these changes in aerosol composition.

Figure 22 shows boxplots of seasonal SSA over Europe. Similar to North America, Europe exhibits clear seasonal variations in SSA, with the lowest values observed during the summer



(JJA) and higher values in the winter (DJF). The lower SSA values in summer can be attributed to increased levels of absorbing aerosols from sources such as industrial emissions, transportation, and biomass burning. The CERES-MODIS data shows more pronounced variability and a wider range of SSA values compared to OMI, particularly during the summer and autumn (SON). This indicates that CERES-MODIS captures the seasonal dynamics of aerosol properties more effectively. The boxplots also reveal outliers, reflecting specific pollution events or variations in aerosol loading.

CERES-MODIS data consistently shows greater sensitivity and variability in SSA values compared to OMI in both regions. Presence of biomass burning aerosols during JJA seasons from forest fires are also captured by CERES-MODIS with lower SSA values.

## 5.2 Polluted land

The Indo-Gangetic Plain and Eastern China are highly suitable for studying polluted regions due to their significant aerosol loading. These areas are characterized by dense populations, extensive industrial activity, thermal power plants and high levels of agricultural burning, which collectively contribute to elevated concentrations of aerosols in the atmosphere (Jethva et al., 2005; Satheesh et al., 2008; Babu et al., 2008; Babu et al., 2016; Satheesh et al., 2017; Gogoi et al., 2020). The Indo-Gangetic Plain, experiences severe air pollution from both biomass burning and industrial emissions. Similarly, Eastern China, with its rapid industrialization and urbanization, faces high aerosol concentrations primarily from coal combustion and vehicular emissions.



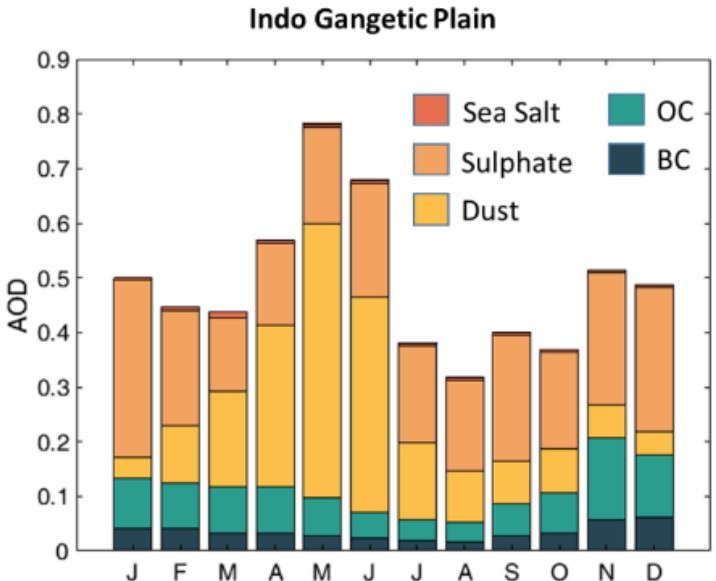

**Fig 23** Variations in monthly mean optical depth of various aerosol types over IGP.

Figure 23 illustrates the monthly variations in AOD for different aerosol types over the Indo-Gangetic Plain (IGP). The data reveals significant seasonal fluctuations, with the highest AOD

5    values occurring from May to June. Dust levels peak significantly during these summer months in the IGP, due to the strong winds of the pre-monsoon season that effectively carry dust from the Thar Desert into nearby regions (Moorthy et al., 2007; Banerjee et al., 2021). Organic Carbon and Black Carbon also show notable contributions, reflecting biomass burning and industrial emissions. Sulphates contribute consistently throughout the year, with slight

10    increases during the winter months. Sea salt aerosols are present in minimal amounts, indicating limited influence from marine sources. Analysis shows that the AOD pattern in the IGP is characterized by high dust levels in the pre-monsoon period and significant contributions from anthropogenic aerosols year-round.

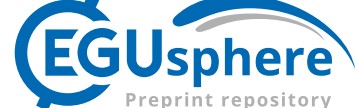

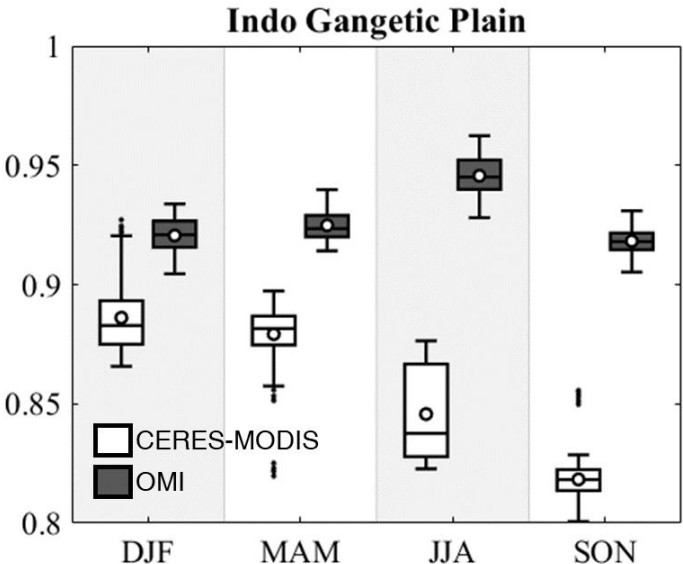

**Fig 24**. Boxplots indicating seasonal SSA over Eastern China. The white boxes represent SSA values retrieved by CERES-MODIS, and the black boxes represent SSA values retrieved by OMI. Each boxplot displays the median (central line), the IQR from the 25th percentile (bottom edge of the box) to the 75th percentile (top edge of the box), and the whiskers, which extend to 1.5 times the IQR or to the maximum and minimum values within this range. Outliers beyond the whiskers are plotted as individual points.

Figure 24 presents boxplots showing the seasonal SSA over the Indo-Gangetic Plain, comparing values retrieved by CERES-MODIS (white boxes) and OMI (black boxes). Extensive cloud coverage during monsoon period can generate outlier low SSA values in CERES-MODIS. The lower values are observed during the pre-monsoon (MAM) aligns with the high dust AOD observed in the previous figure (Fig 23).



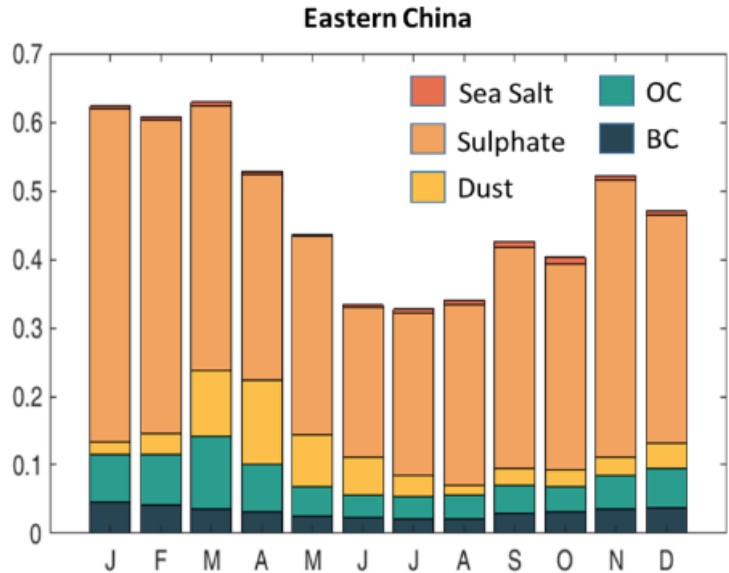

**Fig 25** Variations in monthly mean optical depth of various aerosol types over Eastern China

Figure 25 shows the monthly variations in AOD for various aerosol types over Eastern China. Sulphates are the predominant aerosol type throughout the year, with consistently high levels from January to December. This reflects the extensive industrial activities and coal combustion in the region, contributing to high sulphate emissions. Organic Carbon and Black Carbon also show substantial contributions, particularly during the winter months, likely due to residential heating and biomass burning. Dust aerosols exhibit noticeable peaks in April and October, suggesting seasonal dust storms and agricultural activities as significant sources. Sea salt aerosols remain relatively constant and low, indicating a limited marine influence.

Figure 26 shows boxplots of seasonal SSA over Eastern China, with CERES-MODIS values represented by white boxes and OMI values by black boxes. Similar to the IGP, the lowest values are observed during the summer (JJA) and autumn (SON) periods. These lower SSA values correspond to the higher levels of absorbing aerosols, particularly sulphates and black carbon, as indicated in the previous figure (Fig 25). The consistently high sulphate levels





throughout the year reflect the region's extensive industrial activities and coal combustion, contributing to the lower SSA values.

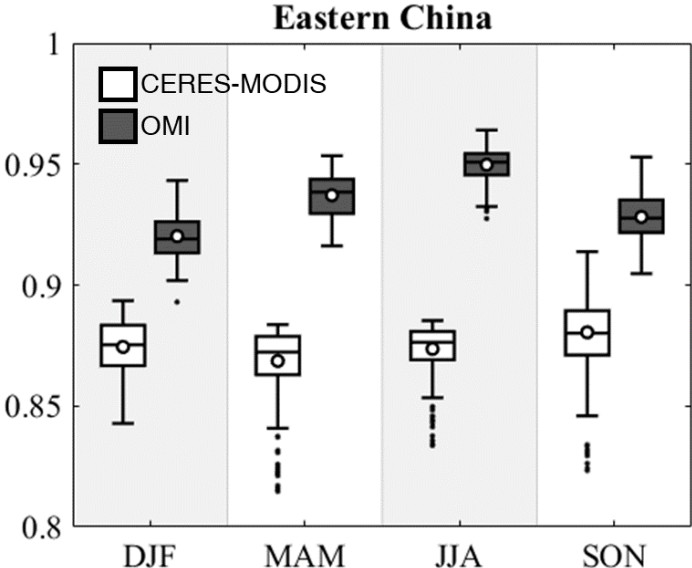

5    **Fig 26**. Boxplots indicating seasonal SSA over Eastern China. The white boxes represent SSA values retrieved by CERES-MODIS, and the black boxes represent SSA values retrieved by OMI. Each boxplot displays the median (central line), the IQR from the 25th percentile (bottom edge of the box) to the 75th percentile (top edge of the box), and the whiskers, which extend to 1.5 times the IQR or to the maximum and minimum values within this range. Outliers beyond

10    the whiskers are plotted as individual points.

# 6 Desert

The Sahara, the largest desert in the world, spans approximately 9.2 million squares kilometres across North Africa. It is a major source of atmospheric dust, which is transported across vast





distances by strong winds (Goudie et al., 2001; Middleton and Goudie., 2001; Tanré et al., 2003). This dust plays a crucial role in various environmental and climatic processes, such as fertilizing the Amazon rainforest with nutrients carried across the Atlantic Ocean (Swap et al., 1992; Koren et al., 2006; Rizzolo et al., 2017; Nogueira et al., 2021). The Sahara's dust also

impacts air quality and weather patterns in regions as far as the Caribbean and the Americas (Petit et al., 2005; Garrison et al., 2006; Rodríguez et al., 2024; Vallès-Casanova et al., 2025)

Figure 27 illustrates the variations in monthly mean AOD for different aerosol types over the Sahara. The data clearly shows that dust is the dominant aerosol type throughout the year, with significant seasonal fluctuations. The highest AOD values are observed from May to August,

peaking in June and July. This period corresponds to the summer months when strong winds and dust storms are most prevalent, lifting large quantities of dust into the atmosphere. Sulphates, organic carbon and black carbon contribute minimally to the overall AOD, with slight increases observed during the peak dust months. Sea salt aerosols are present in negligible amounts, indicating minimal marine influence. Overall, it demonstrates the

dominance of dust aerosols over the Sahara throughout the year. The substantial seasonal variability in AOD is primarily driven by changes in wind patterns and atmospheric stability, highlighting the strong influence of meteorological conditions on dust emissions and transport from this region.





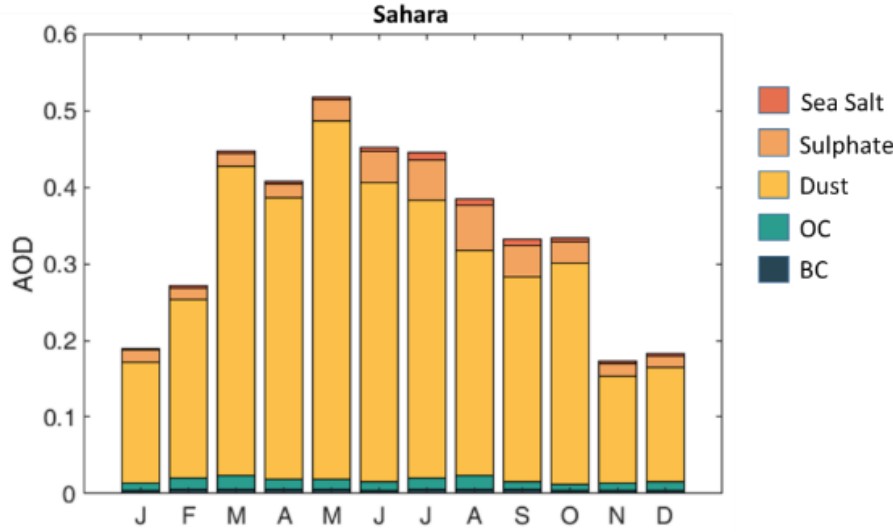

**Fig 27** Variations in monthly mean optical depth of various aerosol types over Sahara.

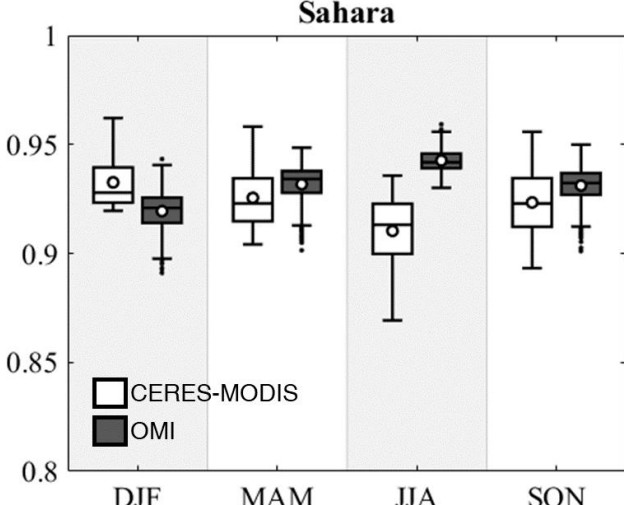

5    **Fig 28**. Boxplots indicating seasonal SSA over Sahara. The white boxes represent SSA values

retrieved by CERES-MODIS, and the black boxes represent SSA values retrieved by OMI.

Each boxplot displays the median (central line), the IQR from the 25th percentile (bottom edge





of the box) to the 75th percentile (top edge of the box), and the whiskers, which extend to 1.5 times the IQR or to the maximum and minimum values within this range. Outliers beyond the whiskers are plotted as individual points.

Figure 28 presents boxplots indicating SSA over the Sahara, comparing values retrieved by
CERES-MODIS (white boxes) and OMI (black boxes). The data reveals significant seasonal variations in SSA, with lower values observed during the summer (JJA) months. This decrease in SSA during the summer aligns with the peak dust AOD observed in Fig 27, as the high concentration of dust aerosols, which have lower SSA values, reduces the overall SSA. The CERES-MODIS data shows greater variability and generally lower SSA values compared to
OMI, particularly during the summer and autumn (SON) seasons.

Both CERES-MODIS and OMI perform similar in retrieving SSA for dust aerosols, as evidenced by the figures. The seasonal boxplots (Fig 28) show that both instruments capture the expected seasonal variations in SSA over the Sahara, with lower values during the summer months (JJA) when dust concentrations are highest. Although CERES-MODIS exhibits slightly
greater variability and generally lower SSA values, both datasets reflect the dominance of dust aerosols accurately. This consistency between CERES-MODIS and OMI indicates that both instruments are capable of reliably monitoring SSA variations due to dust aerosols, despite their different retrieval algorithms and spectral sensitivities.

## 7 Summary and Conclusion

Performance evaluation of CERES-MODIS and OMI was conducted in SSA retrieval across various regions, with a focus on biomass burning areas, oceanic regions, and desert areas. The regional analysis aimed to provide insights into the strengths and weaknesses of each dataset in capturing aerosol properties under diverse environmental conditions.



The analysis of the Amazon and South African forests demonstrated significant seasonal variations in aerosol optical depth (AOD) and SSA. In the Amazon, CERES-MODIS SSA showed a strong negative correlation with fire counts, having a regression slope of -0.0016 and an $R^2$ value of 0.711. This indicates that increased fire activity, primarily during the dry season (August to October), leads to higher concentrations of absorbing aerosols like black carbon, thereby lowering SSA. In contrast, OMI data showed a much weaker correlation ($R^2 = 0.0197$). Similarly, for the South African forest, CERES-MODIS SSA also displayed a strong negative correlation with fire counts ($R^2 = 0.824$), whereas OMI data showed a weaker correlation ($R^2 = 0.42$). Thus CERES-MODIS exhibits higher sensitivity to absorption by biomass burning aerosols compared to OMI.

Analysis of aerosol loading and SSA variations across different continental regions, including North America, Europe, the Indo-Gangetic Plain, and Eastern China was performed. The study highlights monthly variation in aerosol types, such as dust, organic carbon, black carbon and sulphate. The seasonal SSA values retrieved by CERES-MODIS and OMI are compared, to evaluate the effectiveness of these algorithms in capturing aerosol absorption and scattering. Over relatively pristine regions, both OMI and CERES-MODIS captures similar SSA values with lower values by OMI and more variations captured by CERES-MODIS. Biomass burning aerosols occurring seasonally over clean land is captured well by CERES-MODIS whereas, over highly polluted land regions, OMI consistently shows higher SSA values. CERES-MODIS retrieved values are closer to ground station measurements and seasonal variations.

The South Pacific Ocean, representing a relatively clean oceanic region, showed high SSA values throughout the year, consistent with the dominance of scattering aerosols like sea salt. Both CERES-MODIS and OMI captured these variations, but OMI showed very low SSA values even while sea salt aerosols were dominant. This could be due to extensive cloud cover affecting the SSA retrievals. In contrast, the Arabian Sea and Bay of Bengal, representing

polluted oceanic regions, exhibited significant seasonal fluctuations in AOD and SSA. High AOD values during the monsoon season in the Arabian Sea and pre-monsoon season in the Bay of Bengal were driven by sea salt, dust, and anthropogenic aerosols. CERES-MODIS data indicated more pronounced seasonal variations in SSA, highlighting its sensitivity to changes in aerosol composition and loading in polluted oceanic regions.

The Sahara Desert, a major source of atmospheric dust, displayed significant seasonal variations in AOD and SSA. Peak AOD values occurred from May to August, corresponding to intense dust storms. Both CERES-MODIS and OMI effectively captured the seasonal SSA variations, with lower values during the summer months due to higher dust concentrations. Both datasets exhibited similar performance in detecting dust aerosols.

Detailed analysis of CERES-MODIS, and OMI observations across several regions across the world, shows that it effectively captures various fire events, biomass burning activities, and dust episodes when compared to OMI while retrieving SSA in the visible region. The regional studies highlight the strengths and weaknesses of each algorithm in various aerosol environments, providing detailed insights into their performance at specific locations. In conclusion, the CERES-MODIS dataset offers greater accuracy in capturing SSA values in the visible wavelengths. These distinctions allow researchers to choose the appropriate dataset based on their specific needs for temporal resolution and wavelength.

**Data Availability**

MODIS and CERES data used in this study are available at https://asdc.larc.nasa.gov/

**Author Contributions**

AD conceptualized the comparative performance evaluation and conducted the analysis with input from SKS and JS. AD drafted the manuscript, with revisions provided by SKS and JS.



**Competing interests**

The authors declare they have no conflict of interest.

**Acknowledgment**

The authors gratefully acknowledge the Atmospheric Science Data Center (ASDC) at NASA's

Earth Observing System Data and Information System (EOSDIS) Distributed Active Archive

Centers (DAACs) for providing MODIS, OMI, and CERES data products used in this study.

In addition, one of the authors (S. K. Satheesh) acknowledges the JC Bose Fellowship awarded

to him by SERB-Department of Science and Technology, New Delhi.

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
