# Peer review of "Performance comparison between CERES-MODIS and OMI in retrieving SSA across diverse aerosol regimes"

_EGUsphere, 2025_

## Referee Comment (RC3)

Review of manuscript 'Performance comparison between CERES-MODIS and OMI in retrieving SSA across diverse aerosol regimes' by Archana Devi, Sreedharan K Satheesh, and Jayaraman Srinivasan, submitted for publication in AMT.

In this work the authors attempt to examine the performance of two satellite-based approaches to retrieve aerosol single scattering albedo (SSA). The satellite SSA products under evaluation are the one obtained from the combined use of CERES-MODIS observations developed by the manuscript authors, and the SSA product derived from radiance observations by the Ozone Monitoring (OMI) on the Aura NASA satellite. Because there are two independent OMI SSA products derived from two different algorithms referred to as OMAERO and OMAERUV (Torres et al., 2007), in the manuscript under review it is not clearly stated which OMI SSA product was used by the authors in their analysis.

The evaluation approach of the SSA retrieval techniques over major biomass burning regions of the world documented in this article uses the observed correlation between the satellite retrieved SSA parameters and fire counts. As the number of fires increases the aerosol load, i.e., aerosol optical depth, will also increase. The author's SSA evaluation approach assumes that the number of fires per unit area should be inversely correlated to SSA magnitude. They explicitly state their expectation that higher fire activity (i.e., increased fire counts) leads to aerosols with higher absorptivity, and thus lower SSA.

Increased number of fires per unit area will certainly result in increased values of both extinction optical depth and absorption optical depth but not an increase in aerosol absorptivity associated with decreased SSA. The SSA is an intensive aerosol property. It depends solely on the aerosol composition and not on the AOD magnitude. Because aerosol composition does not change as the number of fires increases, observing increased aerosol absorptivity with increased fire counts is a flawed expectation. Thus, the retrieved SSA should show no discernable correlation with the aerosol amount or with fire counts. Burning of the same type of vegetation, regardless of the amount burned, should yield similar SSA values which consistent with the results of OMI-SSA versus fire counts discussed in the manuscript.

In the opinion of this reviewer, expecting a lower SSA due to occurrence of higher AOD because of multiple fires is an ill-conceived assumption. I therefore do not recommend the publication of this article.

I suggest the authors carry out a validation of the CERES-MODIS SSA product by a direct comparison to SSA inversions by the Aerosol Robotic Network (AERONET) as it has been done in the evaluation of other satellite SSA products.

| I would have additional comments on this manuscript but since I anticipate a total rewriting of this work I do not think they are necessary at this time. |  |
|-----------------------------------------------------------------------------------------------------------------------------------------------------------|--|
|                                                                                                                                                           |  |
|                                                                                                                                                           |  |
|                                                                                                                                                           |  |
|                                                                                                                                                           |  |
|                                                                                                                                                           |  |
|                                                                                                                                                           |  |
|                                                                                                                                                           |  |
|                                                                                                                                                           |  |
|                                                                                                                                                           |  |
|                                                                                                                                                           |  |
|                                                                                                                                                           |  |
|                                                                                                                                                           |  |
|                                                                                                                                                           |  |
|                                                                                                                                                           |  |
|                                                                                                                                                           |  |
|                                                                                                                                                           |  |
|                                                                                                                                                           |  |
|                                                                                                                                                           |  |
|                                                                                                                                                           |  |